# Contextual and psychosocial factors influencing caregiver safe disposal of child feces and child latrine training in rural Odisha, India

**Gloria D. Sclar** [1,2]*, **Valerie Bauza**[1], **Alokananda Bisoyi**[3], **Thomas F. Clasen**[1], **Hans-Joachim Mosler**[4]

**1** Gangarosa Department of Environmental Health, Rollins School of Public Health, Emory University, Atlanta, Georgia, United States of America, **2** Department of Psychology, University of Zürich, Zürich, Switzerland, **3** Independent Consultant, Berhampur, Odisha, India, **4** RanasMosler, Zürich, Switzerland

* gloria.sclar@emory.edu

**Data Availability Statement:** The data used in this study are available in the Dataverse open data repository (https://doi.org/10.15139/S3/OOSYWX).

## Abstract

Child feces are an important source of fecal exposure in household environments. Typically, one of two behaviors is necessary to mitigate this risk: either caregivers dispose of their children's feces into a latrine or children learn how to use a latrine. Although past studies have examined factors associated with these two behaviors collectively (i.e. "safe disposal"), there is a need to separately analyze these distinctive practices to better inform programming. This study aims to quantitatively examine contextual and psychosocial factors influencing caregiver safe disposal and, separately, child latrine training. We surveyed 791 primary female caregivers, who reported on 906 children <5 years old, across 74 villages in rural Odisha, India. At their last defecation event, 38% of children used the latrine and another 10% had their feces safely disposed of into the latrine. Since caregiver safe disposal was rare, we instead assessed safe disposal *intention*. We used linear regression and multi-level mixed effects models to examine contextual and psychosocial factors. For contextual factors, we found caregivers had stronger safe disposal intention when they came from wealthier households and had greater informational support, but weaker intention when their latrine was near the household. Caregivers more intensely practiced latrine training with their child when they themselves used the latrine for defecation, the latrine was fully intact, and they had greater instrumental support. For psychosocial factors, caregivers had stronger safe disposal intention when their households expected them to practice safe disposal, they felt strongly committed to the behavior, and had a plan for what to do when faced with a water shortage. Caregivers more intensely taught their child how to use the latrine when they believed their child was at risk of becoming sick if they practiced open defecation (OD); viewed child OD as unbeneficial; liked teaching their child; personally felt it was important for the child's father to help; felt confident in their ability to teach their child; and had greater action control over their training practice. Interestingly, caregivers put less effort into latrine training when they felt more concerned for their child's safety when the child defecated outside. These findings underscore the critical need to separately assess unique child

**Funding:** TC received an award to fund this study from the Bill & Melinda Gates Foundation (https://www.gatesfoundation.org/; grant number = INV-008967). VB was funded in-part by the National Institute of Environmental Health Sciences, USA (https://www.niehs.nih.gov/; grant number = T32ES012870). The funders had no role in study design, data collection and analysis, decision to publish, or preparation of the manuscript.

**Competing interests:** The authors have declared that no competing interests exist.

feces management (CFM) practices and also provide a road map for practitioners on the types of behavior change strategies to consider in their CFM programming.

## Introduction

Child feces are an important source of fecal exposure in household environments and pose a risk to health. Children under five often defecate in or near the household, with research showing young children's feces are a more common source of fecal contamination than other household members [1, 2]. Evidence also suggests child feces may contain more pathogens than adult feces [3]. Young children themselves are especially at risk of exposure as they actively engage with their surroundings and practice exploratory behaviors such as mouthing [4, 5]. For these reasons, disposal of child feces anywhere other than an improved latrine is considered unsafe [6]. Yet, safe child feces disposal and child latrine use are often not practiced. An analysis of 34 low- and middle-income countries estimated 50.6% of households do not safely dispose of their child's feces into a latrine or have the child use the latrine [7]. While the world has seen great progress in increasing sanitation access and reducing adult open defecation [8], poor child feces management (CFM) practices remain largely ignored.

In India, safe child feces disposal and child latrine use are especially uncommon practices, despite recent gains in sanitation coverage. Based on the latest National Family Health Survey, only 36% of Indian households with small children disposed of their child's feces into a latrine or their child used the latrine, even though 61% of households had access to a latrine [9]. The state of Odisha, with 35% latrine coverage, reported the lowest rate with only 13% of households disposing of their child's feces into a latrine or their child using the latrine [9]. Majorin et al. [11] examined the CFM practices among households in rural Odisha and found that instead of practicing disposal of child feces into a latrine or child latrine use, the majority of children <5 defecated on the ground inside the household or around the household compound (64.8%) and their feces ended up in the household solid waste pile (60.7%). In another cross-sectional study among urban slums in Odisha, most children <5 defecated directly on the ground and their feces were disposed of in the household garbage or in a canal/drain [10]. In both the rural and urban settings, only about one fifth of child feces ended up in the latrine (22% and 25%, respectively) with the majority of this from older children actually defecating in a latrine [10, 11]. There is a clear need to better understand the factors at play for caregiver safe disposal and child latrine use in order to design effective behavior change programming and ultimately improve health.

Much of the CFM literature to date focuses on the broad behavioral term of "safe disposal" defined by the WHO/UNICEF Joint Monitoring Programme [6]. However, this term was derived through a health and measurement lens and in actuality encompasses two distinct "safe" behaviors: child uses the latrine *or* caregiver puts/rinses the child's feces into the latrine [6]. From a behavioral lens, it is imperative to disentangle this term and examine the two behaviors separately. While both behaviors are important for reducing fecal exposure, there are likely different factors influencing each that must be uncovered in order to actualize behavior change and subsequent health gains. Here we will use the phrase "caregiver safe disposal" to describe the behavior of the caregiver herself disposing of her child's feces into a latrine. This behavior consists of a singular actor—the caregiver. In contrast, "child latrine use" refers to the child themselves using the latrine for defecation and it is the child who is the behavioral actor. An area severely underexplored in the CFM literature is the preceding behavior of "child latrine training." In order for a child to use the latrine, they must first undergo a

learning behavioral process whereby the caregiver teaches the child, to at least some degree of intensity, how to use the latrine. This behavior of "child latrine training" is a *dyad* behavior that involves both the child *and* the caregiver. In this study we separately assess caregiver safe disposal and child latrine training, with both examinations done from the perspective of the caregiver.

Health behavior change theories outline a variety of factors one must explore in order to diagnose and better understand a behavior. Several theories have been applied to water, sanitation and hygiene (WASH) behaviors such as the integrated behavioral model for WASH (IBM-WASH), behavior centered design (BCD), and RANAS model [12–14]. All highlight the need to assess both contextual and psychosocial factors related to behaviors. Contextual factors include personal characteristics and aspects of the environment which may influence behavior. The environment is often viewed as physical elements that enable or impede behavior but it can also include social elements; for example, the availability of social support. Contextual factors are typically not the focus of behavioral interventions, but an understanding of these factors can better tailor strategies to certain participants and also reveal external barriers to behavioral performance. Psychosocial factors encompass the cognitive elements that influence behavior and have potential for change, such as attitudinal beliefs and normative expectations. The RANAS model in particular outlines five specific psychosocial factors which make-up the theory's acronym: Risks, Attitudes, Norms, Abilities, and Self-regulation [14].

The past decade has seen a surge in studies that examine contextual factors associated with safe disposal of child feces. Many of these studies analyzed Demographic Health Survey (DHS) data and study settings included India, Bangladesh, Cambodia, and sub-Saharan Africa. Taken together, this body of research has uncovered several common contextual factors related to household, caregiver, and WASH characteristics. Studies show that households are more likely to practice safe child feces disposal when they are in urban settings [15–18], wealthier [15–23], belong to certain religions [10, 16, 17, 23], and when household members practice latrine use [10, 22, 24–26]. At the caregiver level, studies found older mothers [17, 18, 23, 26] who are literate or have higher levels of education [10, 15, 16, 19, 22, 23] and greater exposure to media [16, 17, 21] are more likely to practice safe disposal. For the WASH environment, households with a latrine or water within their household compound; access to an improved latrine or water source in particular; or water was observed at a place for handwashing, were also more likely to safely dispose of their child's feces [10, 15–17, 19, 21, 23, 25, 27]. However, some contextual factors identified as associated with "safe disposal" may in fact be associated with only one of the two behaviors this broader term covers. For example, many studies found older or ambulatory children are more likely to have their feces safely disposed [10, 11, 15, 17, 18, 21, 24–26], but these characteristics may be specifically associated with child latrine use rather than caregiver disposal of child feces into a latrine.

Compared to contextual factors, there is less research on the psychosocial factors that influence safe CFM practices; still, some shared themes have emerged from qualitative studies. A common finding is a lack of risk perception. Some caregivers do not perceive infant feces to be harmful and as such, do not think they need to be disposed of in a latrine. This low risk perception is attributed to infants' breastmilk-only diet which makes their feces not smell and appear light in color. However, the feces of older children are often perceived as a health risk because when children transition to solid foods their feces take on the more typical odor and appearance of adult feces [20, 28–31]. Consequently, another common finding is the negative attitudes and social norms caregivers experience when their child defecates around the household environment. Many studies documented the disgust caregivers feel towards seeing their child's feces and also the social judgement they face, or anticipate, from having a dirty home [28, 31–33].

A few qualitative studies specifically explored caregivers' experiences with potty training and child latrine use. The studies uncovered several psychosocial factors that acted as barriers to behavioral performance. When it came to potty or toilet training, some caregivers held negative attitudes towards the practice because of its many perceived costs: training is difficult, it takes time, and there are other competing tasks that the caregiver needs to prioritize [26, 34]. The latrine itself is also sometimes perceived as a dangerous environment for small children. Caregivers fear their child might slip or fall into the squat hole and injure themselves [26, 34]. The latrine is also seen as a contaminated place that may make their small children sick [34].

This study aims to quantitatively examine contextual and psychosocial factors influencing caregiver safe disposal and, separately, child latrine training among households in rural Odisha, India. To our knowledge, this is the first quantitative study that assesses behavioral factors for these distinct CFM practices. We first assessed the prevalence of caregiver safe disposal and child latrine use, stratified by child age group, to understand the current state of these behaviors. We then examined personal, physical, and social contextual factors associated with caregiver safe disposal intention and child latrine training. Finally, we applied the RANAS model to more deeply examine psychosocial factors for each behavior. Findings from the study helped inform the design of a behavior change intervention for the Odisha-based NGO Gram Vikas.

## Methods

### Setting and participants

This study used baseline survey data for a randomized controlled trial that engaged 74 villages in Ganjam and Gajapati districts in the state of Odisha. Both districts are mostly rural with agriculture as the primary occupation, but the districts differ in their geography and demography. Ganjam covers a varied geography of hills, valleys, coastal plains and tableland and has a predominantly Hindu population while Gajapati is hilly with a more substantial Christian and Scheduled Tribe population. The trial enrolled 49 villages in Ganjam and 25 villages in Gajapati.

The trial villages were randomly selected from a list of villages that had previously participated in a community-based water and sanitation intervention by Gram Vikas, known as "MANTRA." The intervention involved construction of twin-pit pour flush latrines with attached bathing room and piped water supply. This ensured most households already had an enabling environment to practice caregiver safe disposal and child latrine use. All households with a child <5 years old and a latrine were eligible to participate in the baseline survey. The target participant was the primary caregiver of the child <5 years old but if they were not available then the secondary caregiver of the child was asked to participate. In most cases the participant was the mother of the child but in some cases they were the father, grandmother, grandfather, aunt or another family member. Additional details about the trial study design, including sample size calculation to determine number of clusters (i.e. villages), are provided in Sclar, et al. [35].

### Sample

We restricted the analysis to only female primary caregivers. Secondary caregivers were excluded because their involvement and role in child feces management is more varied compared to primary caregivers. Male primary caregivers were excluded since so few were surveyed, making it difficult to examine gender differences.

## Data collection procedure

Data collection took place during the winter season between December 2019 to February 2020. The data collection team consisted of 11 surveyors (10 women, 1 man) and a research manager (author AB), all of whom were fluent in the local language Odia. To ensure consistent data collection and accuracy of the survey tool, the team underwent a week-long training followed by several days of pilot-testing in non-trial villages. The survey was translated from English to Odia and checked by the data collection team during training, and further refined based on pilot-testing. Survey responses were recorded on a secure Android mobile using the open source application Open Data Kit (ODK) Collect (https://opendatakit.org/).

In each study village, the data collection team attempted to survey all eligible households in the village. Once a household was confirmed eligible, the data collector informed the target participant (i.e. primary or secondary caregiver) about the study and obtained their oral consent to proceed. The survey took approximately 45 to 60 minutes to complete. The research manager monitored the team throughout data collection and conducted unannounced observations of surveys as a quality control measure.

## Survey and measures

The survey consisted of six sub-questionnaires: participant and household demographics, caregiver perceived stress, child demographics and child feces management practices, psychosocial factors related to latrine training/disposal, received social support with latrine training/disposal, and characteristics of household water and sanitation facilities. The survey was followed by a structured spot-check of the household's latrine to assess condition. There were two versions of the psychosocial and social support sub-questionnaires of which caregivers only received one. If the caregiver perceived her child too young to learn to use the latrine, she answered the child feces disposal versions. If the caregiver was currently teaching her child to use the latrine or perceived her child to be old enough, she answered the latrine training versions.

**CFM practices.** Caregivers were asked a series of questions about what happened *the last time* their child defecated in order to measure their CFM practice. If the caregiver had more than one child <5 years old, then the questions were repeated for each child. Additionally, if the caregiver did not know about the last time, which was rare (n = 7; <1%), then they were asked about what happens "usually." Caregivers were first asked, *"The last time the child defecated, where did they defecate?"* If the caregiver reported "in the latrine" then this was categorized as child latrine use. If not, a follow-up question was asked: *"Where were the child's feces disposed?"* If the caregiver reported "in the latrine" then this was categorized as caregiver safe disposal. All other responses were categorized as unsafe disposal.

**Safe disposal intention.** Since caregiver safe disposal was rare (10%, see Table 3), we decided to examine caregiver's safe disposal intention. Behavioral intention indicates how motivated and willing a person is to perform a behavior, and is a necessary precursor for behavioral action [36]. Safe disposal intention was measured by a self-reported question in the psychosocial factors sub-questionnaire—*"How strongly do you intend to always dispose of your child's feces into the latrine?"*—with a 5-point Likert scale response from 1 (not at all) to 5 (very strongly).

**Latrine training intensity.** Latrine training intensity, defined as how much effort a caregiver was giving to teaching her child how to use the latrine, was measured based on two self-reported questions. First, caregivers were asked *"Are you currently teaching your child how to use the latrine for defecation?"* with different clarifying response options (yes, no—my child is not old enough, no—my child already knows how to use the latrine, no—though my child is

old enough). This question assessed whether the caregiver herself perceived she had initiated the latrine training process with her child. If the caregiver reported "yes," then a follow-up was asked—*"During the last week, when your child needed to defecate, how often did you take your child to the latrine and teach them how to use it?"*—with a 5-point Likert scale response from 1 ((almost) never (0%)) to 5 ((almost) always (100%)). Latrine training intensity was thus measured on a 0 to 5 scale based on the response to the second question and with 0 assigned to caregivers who reported "no-though my child is old enough" to the first question. Caregivers who perceived their child was not yet old enough to learn how to use the latrine or who reported their child already knew how to use the latrine were not assigned a latrine training intensity value. If the caregiver had multiple children <5 years old who she was teaching how to use the latrine or she perceived old enough to learn, then responses for the youngest child were used in the analysis.

**Contextual factors.** The contextual factors examined included personal, social, and physical characteristics: personal characteristics of the caregiver, child and household; characteristics of the amount and type of social support that the caregiver received; and physical characteristics of the household's water and latrine infrastructure.

Caregiver characteristics were self-reported and included age, number of years of education, whether or not the caregiver was employed/self-employed, and whether or not the caregiver used the latrine the last time she defecated.

Child characteristics included the child's age, sex, and developmental abilities. For developmental abilities, the caregiver was asked to report "yes" (1) or "no" (0) if her child was doing any of the following: walk on their own, squat on their own, speak in full sentences, follow directions, and eat adult foods. The developmental abilities were analyzed individually as dichotomous variables. However, in the latrine training intensity analysis, since almost all children were able to walk (97%) and ate adult foods (98%) these were excluded from the analysis. In the safe disposal intention analysis, only the ability to walk was examined. For caregivers with multiple children <5 years old, the characteristics of the youngest child were included in the analysis.

Household characteristics included religion, caste group, whether or not the household had multiple children <5 years old, the number of household members that helped with childcare, and wealth. Household wealth was measured by constructing an asset index (scooter/motorcycle, television, telephone [landline or mobile], refrigerator, mattress, cot, chair, table, sewing machine, pressure cooker, watch/clock, electric fan, water pump, animal drawn cart, thresher, tractor, electricity, livestock) and using polychloric principles component analysis to categorize households into five wealth quintiles (quintile 1 –least wealthy; quintile 5 –most wealthy) [37].

The amount of social support the caregiver received with latrine training/disposal was measured using the received social support sub-questionnaire. The questionnaire examined three types of support—emotional, instrumental, and informational. The items were adapted from several validated metrics [38–42] with several items examining one type of social support. In each item, the caregiver was asked about a specific supportive act she experienced in the last week with a 6-point Likert scale response from 1 (completely disagree) to 6 (completely agree). Many of the supportive acts used in the items came from real examples provided by caregivers in prior qualitative research. A score was constructed for each of the three types of social support by taking the average of their relevant items. A higher score indicates greater received support. Example items and the Cronbach's alpha internal reliability coefficient for each constructed score are presented in Tables 1 and 2.

Household water and latrine characteristics were either reported by the caregiver or observed by the surveyor. Piped water access was measured as the number of hours in the last

**Table 1. Description of psychosocial and social support factors included in the models for caregiver safe disposal.**

| | Factors | Example item | No. of items (α)* |
|---|---|---|---|
| | *Caregiver safe disposal* | | |
| *RISKS* | *Perceived vulnerability* | How high or low is the risk of you becoming sick if you accidentally eat food contaminated with your child's feces? | 1 |
| *ATTITUDES* | *Positive attitudes (towards safe disposal)* | How easy is it for you to dispose of your child's feces in the latrine? | 3 (0.69) |
| | *Negative attitudes (towards unsafe disposal)* | How disgusted do you feel when you see your child's feces in the back of your house? | 3 (0.63) |
| *NORMS* | *Personal norm* | How important is it to you that your child's feces are disposed of in the latrine? | 1 |
| | *Personal norm (motherhood)* | I believe a good mother disposes of her child's feces into the latrine. | 1 |
| | *Village descriptive norm* | Among the people you know in this village, how many dispose of their child's feces into the latrine? | 1 |
| | *Household injunctive norm* | Other members of your household expect you to dispose of your child's feces into the latrine. | 1 |
| | *Village injunctive norm* | People in this village will scold you if you dispose of your child's feces outside. | 1 |
| *ABILITY* | *Self-efficacy* | How confident are you in your ability to dispose of your child's feces into the latrine? | 3 (0.73) |
| *SELF-REGULATION* | *Barrier planning* | Do you have a plan for how you will dispose of your child's feces into the latrine when there is a water shortage? | 1 |
| | *Commitment* | How committed are you to ALWAYS disposing of your child's feces into the latrine? | 1 |
| *SOCIAL SUPPORT* | *Emotional support* | In the last week, someone comforted me when I was struggling with managing my child's feces. | 6 (0.78) |
| | *Instrumental support* | In the last week, someone helped with the cooking or cleaning for me so I could go manage my child's feces. | 6 (0.82) |
| | *Informational support* | In the last week, someone helped me make a decision about how to properly manage my child's feces. | 3 (0.86) |

*Cronbach's alpha (α) was calculated for factors with multiple items to assess internal reliability of the constructed factor score.

24 hours that caregivers reported they could *not* get water from their private tap. If the caregiver's household did not have piped water (n = 56; 7%), then a value of 24 was assigned. For the household latrine, caregivers reported on the number of pits and whether or not the latrine was currently used for defecation. However, the latter was dropped from the analysis since almost all reported the latrine to be in use (93%). Surveyors observed the location of the latrine ("in/within 50 feet of the household compound" (1) or ">50 feet" (0)) and if it had functional piped water ("tap in latrine and water comes out when tested" (1) or "no tap/water does not come out" (0)). Lastly, we determined if the latrine structure was "fully intact" (1) or "not fully intact" (0) based on surveyor observations. The surveyor recorded "yes" or "no" if the roof, walls, floor, and door of the latrine were intact, if the walls were at least 5 feet high, and if the squat pan could be used. If "yes" was answered for all items then the latrine structure was considered "fully intact." Pit condition was not included because most surveyors recorded pits as "not visible" since they were often below ground.

**Psychosocial factors.** The psychosocial sub-questionnaires were based on the RANAS model and heavily informed by qualitative research. Items discussed both the desired behavior (i.e. disposal of child feces into the latrine or teaching child how to use the latrine) and the undesired behavior (i.e. disposal of child feces outside or child defecating outside). Most items used a 5-point Likert response scale but a few items were dichotomous. The safe disposal questionnaire included a "yes" (1) or "no" (0) item that examined village injunctive norms and an item on barrier planning was turned dichotomous with 0 indicating no plan and 1 indicating a plan. The latrine training questionnaire included one item that examined caregivers' personal

**Table 2. Description of psychosocial and social support factors included in the models for child latrine training.**

| | Factors | Example item | No. of items (α)* |
|---|---|---|---|
| | *Child latrine training* | | |
| **RISKS** | *Perceived vulnerability of child OD* | How high or low is the risk of your child becoming sick from defecating outside? | 1 |
| **ATTITUDES** | *Unbeneficial for child OD* | How beneficial is it for you to let your child defecate outside? | 1 |
| | *Safety concern w/ child OD* | How concerned are you for your child's safety when they go outside for defecation? | 1 |
| | *Difficulty—latrine training* | How difficult is it for you to teach your child how to use the latrine for defecation? | 1 |
| | *Like—latrine training* | How much do you like teaching your child how to use the latrine for defecation? | 1 |
| | *Irritated—latrine training* | How irritated do you feel when you have to stop what you are doing and help your child defecate in the latrine? | 1 |
| | *Proud—latrine training* | How proud do you feel when you are teaching your child how to use the latrine for defecation? | 1 |
| **NORMS** | *Personal norm* | How important is it to you personally to teach your child to use the latrine for defecation? | 1 |
| | *Personal norm (age to train)* | At what age do you think a mother should start teaching her child how to use a latrine for defecation? | 1 |
| | *Personal norm (father's role)* | How important is it to you that your child's father helps teach him/her how to use the latrine for defecation? | 1 |
| | *Village descriptive norm* | Out of the children in this village who are between 2 and 3 years old, how many do you think usually defecate in a latrine? | 1 |
| | *Household injunctive norm* | Others members of your household expect you to teach your child how to use the latrine for defecation. | 1 |
| **ABILITY** | *Self-efficacy* | How confident are you in your ability to continue teaching your child how to use a latrine when your child refuses to use the latrine (for example, child cries or won't enter the latrine)? | 4 (0.75) |
| **SELF-REGULATION** | *Action control* | How often do you take your child to the latrine or remind them to go to the latrine when they indicate they need to defecate? | 1 |
| | *Commitment* | How committed are you to teaching your child how to use a latrine for defecation? | 1 |
| | *Intention* | How strongly do you intend to teach your child how to use the latrine for defecation? | 1 |
| **SOCIAL SUPPORT** | *Emotional support* | In the last week, someone listened to me when I needed to talk about my struggles with teaching my child to defecate in the latrine. | 6 (0.80) |
| | *Instrumental support* | In the last week, someone helped my child defecate in the latrine when I was not available to do it. | 4 (0.76) |
| | *Informational support* | In the last week, someone gave me advice on how to teach my child to defecate in the latrine. | 3 (0.82) |

*Cronbach's alpha (α) was calculated for factors with multiple items to assess internal reliability of the constructed factor score.

beliefs about the age at which mothers should start teaching their child how to use a latrine with different ages as response options. This item was turned dichotomous where 1 indicated the caregiver believed children < = 2 years old should be taught and 0 indicated >2 years old. Some 5-point Likert items were combined into a single constructed factor by averaging the responses. All factors were coded so that a higher value indicated greater favorability of the desired behavior. Example items and the Cronbach's alpha internal reliability coefficient for constructed factors are presented in Tables 1 and 2.

## Data analysis

We used bivariate linear regression models to examine the relationship between each behavioral outcome—safe disposal intention and latrine training intensity—and each contextual and psychosocial factor. We then performed four multivariable regression analyses to examine which contextual and psychosocial factors explain safe disposal intention and which explain latrine training intensity, including only those factors that had significant coefficients in the

bivariate regressions (p-value < 0.05). We calculated the intraclass correlation (ICC) for each behavioral outcome to determine if there was substantial between-subject variance and multi-level modeling was needed. The ICC for safe disposal intention was 0% but for latrine training intensity it was 15.6%. These results indicated that which village the caregiver resided in did not explain any of the variance in safe disposal intention but it did explain a substantial portion for latrine training intensity. Accordingly, for safe disposal intention we ran linear regression models with cluster robust standard errors and for latrine training intensity we ran *multilevel* mixed effects linear regression models with robust standard errors. The interpretation of the coefficients is the same for both types of models. Hausman's test confirmed a mixed effects model (fixed-slope random-intercept) was more efficient than a fixed effects model (fixed-slope fixed-intercept) for latrine training intensity. We did not examine village-level contextual factors in the multilevel model for latrine training intensity because the purpose of this study was to inform the design of a behavior change intervention that was meant to be implemented across the 74 different trial villages and thus not tailored to village context. All variance infla-tion factors (VIFs) were low (< = 2.13), signifying no issues of multicollinearity in the regres-sion models, and distribution of the errors was approximately normal upon visual inspection of histogram and quantile normal plots. Analyses were performed in STATA Version 17.

### Ethics review

This study was reviewed and approved by the Institutional Review Board (IRB) of Emory Uni-versity (IRB00115339) in Georgia, USA and the Independent Ethics Committee at Xavier Uni-versity Bhubaneswar (220519) in Odisha, India. Participants provided their verbal consent prior to engaging in the survey.

## Results

A total of 1033 caregivers met eligibility and were asked to participate in the survey. Among these, 41 (4%) did not consent, 113 (11%) were secondary caregivers, 10 (1%) were male pri-mary caregivers, 42 (4%) had a child that had already completed their latrine training, and 36 (3%) ended the survey early. This resulted in 791 primary female caregivers included in the analysis.

### Descriptive statistics

The personal, social, and physical characteristics of the caregivers and their households are presented in S1 Table.

Among the 791 caregivers, all but six—five grandmothers and one aunt—were the mother of the child. Caregivers ranged in age from 18 to 60 years old but were predominantly between 20 to 30 years old (90%) (M [mean] = 26.78 years, SD [standard deviation] = 5.16). The major-ity of caregivers had a primary or secondary education (74%), although 20% had never attended school, and 57% of caregivers were unemployed. Most caregivers (72%) had used a latrine the last time they defecated.

The household religion was either Hindu (83%) or Christian (17%), but four households practiced another religion or no religion (<1%). Households largely belonged to the Other Backward Castes (37%), Scheduled Tribes (23%), or General castes (18%). Most households had only one child <5 years (81%) and on average two or three household members, excluding the caregiver, provided help with childcare in the past two days (M = 2.69, SD = 1.79).

Out of the 791 caregivers, 299 answered the child feces disposal version of the social support sub-questionnaire and 492 answered the latrine training version. For both samples, the average emotional support scores (disposal: M = 3.30, SD = 1.45; latrine training: M = 3.62, SD = 1.51)

and informational support scores (disposal: M = 3.40, SD = 1.81; latrine training: M = 3.52, SD = 1.76) were medium, indicating caregivers somewhat receive these types of support when it comes to managing their child's feces or teaching their child how to use the latrine. However, the average instrumental social support scores were relatively high (disposal: M = 4.62, SD = 1.32; latrine training: M = 4.51, SD = 1.45), indicating caregivers do typically receive tangible assistance with these childcare practices.

Due to their prior engagement in the MANTRA program, the vast majority of caregivers lived in households with satisfactory water and sanitation facilities. Ninety percent of households had piped water to their home and the majority experienced less than five hours without running water in the past day (50%) (M = 9.97 hours, SD = 10.43). Most household latrines were located in or within 50 feet of the household compound (88%), had a fully intact superstructure with usable squat plate (82%), and functional piped water inside (51%). About two-thirds of the latrines had two pits (68%). Very few households had more than one latrine (4%) or shared their latrine with other households (6%).

## Child defecation and feces disposal practices

The caregivers reported on the CFM practice for 906 children <5 years old (Table 3). There was a roughly equal distribution of children across the age groups from 0 to <60 months and 48% of children were female.

At their last defecation event, 38% of children used the latrine, an additional 10% had their feces safely disposed of into the latrine, and the remaining 52% had their feces unsafely disposed. Among children who had their feces unsafely disposed, the most common practices were to dispose of the feces into an open area away from the household compound (29%), into the backyard of the household (19%), or into the household garbage pile (18%). Among children who used the latrine, almost all (93%) were two years old or older, and this age group was also the least likely to have their feces safely disposed of when defecating outside the latrine (3.1%). Accordingly, among children who did have their feces safely disposed of, albeit this was rare, most (81%) were less than two years old.

We examined the defecation and disposal practices by analysis group: latrine training intensity and safe disposal intention. As expected, for the children of caregivers in the latrine

**Table 3. Child feces management practices the last time the child defecated, stratified by child age.**

| Child Age Group [a] | child latrine use | | caregiver safe disposal | | unsafe disposal [b] | | total | |
|---|---|---|---|---|---|---|---|---|
| | N | % | N | % | N | % | N | % |
| 0 to 7 months | 0 | 0% | 36 | 35% | 66 | 65% | 102 | 11% |
| 8 to 11 months | 1 | 2% | 15 | 26% | 42 | 72% | 58 | 6% |
| 12 to 17 months | 6 | 7% | 8 | 9% | 75 | 84% | 89 | 10% |
| 18 to 23 months | 16 | 15% | 15 | 14% | 76 | 71% | 107 | 12% |
| 24 to 35 months [c] | 80 | 43% | 11 | 6% | 95 | 51% | 186 | 21% |
| 36 to 47 months | 130 | 63% | 3 | 1% | 73 | 35% | 206 | 23% |
| 48 to 59 months | 114 | 72% | 3 | 2% | 41 | 26% | 158 | 17% |
| TOTAL | 347 | 38% | 91 | 10% | 468 | 52% | 906 | |

[a] The younger age groups align with motor development milestones, which can influence a child's defecation practice (0–7 months pre-ambulatory, 8–11 months crawling, 12–17 months walking, 18–23 months able to squat)

[b] Three caregivers said "don't know" for where child's feces were disposed (1 in each age group from 12 months to 35 months) and one caregiver declined to answer (24 to 35 months group); these were all categorized as unsafe disposal unsafe disposal

[c] One caregiver in the 24 to 35 months group buried the child's feces and this was categorized as unsafe disposal

training intensity analysis the majority (57%) used the latrine the last time they defecated. For children of caregivers in the safe disposal intention analysis, only 15% had their feces safely disposed the last time; again, highlighting this is an uncommon behavior in the study population. Children in the latrine training intensity group were on average about three years old (M = 38.87 months, SD = 11.49) while children in the safe disposal intention group were on average about one year old (M = 15.09 months, SD = 10.00). Almost all children in the latrine training intensity group could walk (97%) compared to about half in the safe disposal intention group (54%). Children in the latrine training intensity group could also squat on their own (87%), follow directions (87%), and speak in full sentences (79%).

## Bivariate analysis

In the bivariate regressions, a few contextual factors were significant for both safe disposal intention and latrine training intensity, but many differed (S2 and S3 Tables). For personal characteristics, none of the caregiver variables were significant for safe disposal intention but all were significant for latrine training intensity. In contrast, none of the child variables were significant for latrine training intensity but child age and ability to walk were significant for safe disposal intention. The only household characteristic that was significant for both behavioral outcomes was household wealth. Both outcomes were also significantly associated with informational and instrumental support, but only latrine training intensity was associated with emotional support as well. For WASH characteristics, latrine location was significant for both outcomes but only latrine training intensity was also significantly associated with latrine structure fully intact and hours of piped water supply. Lastly, for the psychosocial factors, both behavioral outcomes had significant associations with at least one or several items belonging to each factor type (i.e. risks, attitudes, norms, ability, self-regulation).

## Contextual factors

**Caregiver safe disposal intention.**   The contextual factors multivariate linear regression model for safe disposal intention identified household wealth (*b* = 0.109; 95% CI 0.003, 0.215), latrine location (*b* = -0.645; 95% CI -1.042, -0.247), and informational support (*b* = 0.098; 95% CI 0.002, 0.194) as significant predictors (Table 4). Caregivers from wealthier households and who received more informational support on CFM had a greater intention to practice safe disposal of their child's feces. Caregivers who had a latrine in or near the household compound (within 50 feet) had weaker safe disposal intention compared to caregivers who had a latrine >50 feet from the household compound. The model explained only 8% of the variance.

**Child latrine training.**   The contextual factors multilevel mixed effects regression model for latrine training intensity identified caregiver latrine use (*b* = 1.307; 95% CI 0.877, 1.737), latrine structure fully intact (*b* = 0.452; 95% CI 0.021, 0.883), and instrumental support (*b* = 0.212; 95% CI 0.075, 0.349) as significant predictors (Table 5). Caregivers who used the latrine the last time they defecated, had a fully intact latrine with usable squat plate, and who received greater instrumental support, were more intensely teaching their child how to use the latrine. While the model explained a modest level of the overall and within village variance (19% and 14%, respectively), it explained a high level of the variance between caregivers living in different villages (48%).

## Psychosocial factors

**Caregiver safe disposal intention.**   The mean values of the psychosocial factors for safe disposal intention were medium to high, except for negative attitudes towards unsafe child feces disposal and village descriptive norm (Table 4). For mean perceived vulnerability,

**Table 4. Multivariate linear regression models for caregiver safe disposal intention.**

**Contextual Factors Model (n = 287; groups = 67)***

| Variable | M+ | SD | b | Robust SE | P | CI (95%) | |
|---|---|---|---|---|---|---|---|
| | | | | | | LL | UL |
| Child age (months) | 15.09 | 10.00 | -0.015 | 0.013 | 0.259 | -0.042 | 0.012 |
| Child is ambulatory | 161 | 54% | -0.122 | 0.263 | 0.643 | -0.648 | 0.403 |
| **Household wealth quintile** | 3.07 | 1.41 | 0.109 | 0.053 | **0.044** | 0.003 | 0.215 |
| **Latrine in/near household (<50ft)** | 263 | 89% | -0.645 | 0.199 | **0.002** | -1.042 | -0.247 |
| Instrumental support | 4.62 | 1.32 | -0.002 | 0.081 | 0.976 | -0.164 | 0.159 |
| **Informational support** | 3.40 | 1.81 | 0.098 | 0.048 | **0.046** | 0.002 | 0.194 |
| constant | | | 4.134 | 0.399 | <0.001 | 3.338 | 4.930 |

**Psychosocial Factors Model (n = 268; groups = 65)****

| Variable | M+ | SD | b | Robust SE | P | CI (95%) | |
|---|---|---|---|---|---|---|---|
| | | | | | | LL | UL |
| Perceived vulnerability | 3.34 | 1.51 | 0.057 | 0.040 | 0.162 | -0.023 | 0.137 |
| Positive attitudes (safe disposal) | 3.65 | 1.06 | 0.080 | 0.083 | 0.344 | -0.087 | 0.246 |
| Negative attitudes (unsafe disposal) | 2.78 | 1.22 | 0.018 | 0.057 | 0.749 | -0.095 | 0.132 |
| Personal norm | 3.85 | 1.21 | 0.041 | 0.059 | 0.488 | -0.077 | 0.160 |
| Personal norm (motherhood) | 4.61 | 0.97 | 0.033 | 0.047 | 0.478 | -0.060 | 0.127 |
| Village descriptive norm | 2.19 | 1.78 | 0.013 | 0.035 | 0.711 | -0.057 | 0.084 |
| **Household injunctive norm** | 3.97 | 1.55 | 0.154 | 0.055 | **0.006** | 0.045 | 0.263 |
| Village injunctive norm | 201 | 67% | 0.181 | 0.130 | 0.168 | -0.078 | 0.439 |
| Self-efficacy | 3.41 | 1.16 | 0.090 | 0.074 | 0.227 | -0.058 | 0.238 |
| **Barrier planning** | 188 | 63% | 0.503 | 0.167 | **0.004** | 0.169 | 0.837 |
| **Commitment** | 3.61 | 1.44 | 0.432 | 0.053 | **<0.001** | 0.327 | 0.537 |
| constant | | | 0.111 | 0.282 | 0.696 | -0.452 | 0.673 |

M = mean; SD = standard deviation; SE = standard error (robust SE adjusted for clustering); CI = confidence interval

+N.B. Refer to S2 Table for the sample size for each specific variable

*$R^2$ = 0.08; robust standard errors are adjusted for clustering at the village-level

**$R^2$ = 0.56; robust standard errors are adjusted for clustering at the village-level

caregivers estimated the risk to their health was moderate if they ate food contaminated with their child's feces (M = 3.34). Similarly, caregivers on average estimated moderate severity if their child became sick with diarrhea (M = 3.52). The mean score for positive attitudes towards safe disposal was relatively high (M = 3.65) while the mean score for negative attitudes towards unsafe disposal was much lower (M = 2.78), indicating caregivers positively viewed safe disposal but also did not hold strong negative attitudes towards unsafe disposal. The personal norm factors showed caregivers viewed safe disposal as important to them and was especially linked to being a good mother (M = 3.85 and M = 4.61, respectively). Caregivers also perceived most of their household members expected them to safely dispose of their child's feces (M = 3.97) and 67% of caregivers believed people in their village would scold them if they disposed of their child's feces outside. However, caregivers perceived only some villagers actually dispose of their child's feces into a latrine (M = 2.19), suggesting a weak descriptive norm for safe disposal at the village level. The mean self-efficacy score (M = 3.14) indicated caregivers felt reasonably confident in their ability to perform safe disposal and 63% of caregivers had a plan for how they would safely dispose of their child's feces when there is a water shortage. Lastly, caregivers on average reported they were committed or quite committed to always safely disposing of their child's feces into the latrine (M = 3.61).

**Table 5. Multilevel mixed effects linear regression models for latrine training intensity.**

**Contextual Factors Model (n = 427; groups = 66)\***

| | M[+] | SD | Robust | | | CI (95%) | |
|---|---|---|---|---|---|---|---|
| | | | b | SE | P | LL | UL |
| Caregiver's age (years) | 27.36 | 5.02 | -0.022 | 0.019 | 0.253 | -0.060 | 0.016 |
| Caregiver years of education | 6.60 | 4.19 | 0.041 | 0.026 | 0.117 | -0.010 | 0.093 |
| Caregiver unemployed | 270 | 55% | -0.052 | 0.185 | 0.778 | -0.415 | 0.311 |
| **Caregiver latrine use** | 389 | 79% | 1.307 | 0.219 | **<0.001** | 0.877 | 1.737 |
| Household wealth quintile | 3.13 | 1.39 | -0.017 | 0.075 | 0.818 | -0.164 | 0.129 |
| Hours without piped water | 10.38 | 10.48 | -0.007 | 0.008 | 0.395 | -0.023 | 0.009 |
| Latrine in/near household (<50ft) | 421 | 86% | 0.176 | 0.245 | 0.473 | -0.305 | 0.657 |
| **Latrine structure fully intact** | 393 | 83% | 0.452 | 0.220 | **0.040** | 0.021 | 0.883 |
| Emotional support | 3.62 | 1.51 | -0.013 | 0.068 | 0.842 | -0.146 | 0.119 |
| **Instrumental support** | 4.51 | 1.45 | 0.212 | 0.070 | **0.002** | 0.075 | 0.349 |
| Informational support | 3.52 | 1.76 | -0.013 | 0.062 | 0.832 | -0.135 | 0.108 |
| constant | | | 1.482 | 0.763 | 0.052 | -0.014 | 2.978 |

**Psychosocial Factors Model (n = 442; groups = 65)\*\***

| | M[+] | SD | Robust | | | CI (95%) | |
|---|---|---|---|---|---|---|---|
| | | | b | SE | P | LL | UL |
| **Perceived vulnerability of child OD** | 3.55 | 1.59 | 0.098 | 0.045 | **0.027** | 0.011 | 0.186 |
| **Unbeneficial for child OD** | 4.52 | 1.11 | 0.137 | 0.062 | **0.027** | 0.015 | 0.258 |
| **Safety concern w/ child OD** | 4.14 | 1.18 | -0.127 | 0.062 | **0.041** | -0.250 | -0.005 |
| Difficulty—latrine training | 4.24 | 1.24 | 0.076 | 0.058 | 0.192 | -0.038 | 0.190 |
| **Like—latrine training** | 4.01 | 1.06 | 0.238 | 0.071 | **0.001** | 0.099 | 0.378 |
| Irritated—latrine training | 4.05 | 1.24 | -0.019 | 0.056 | 0.737 | -0.130 | 0.092 |
| Proud—latrine training | 3.29 | 1.49 | -0.030 | 0.048 | 0.532 | -0.125 | 0.065 |
| Personal norm | 4.06 | 1.08 | -0.065 | 0.071 | 0.362 | -0.204 | 0.074 |
| Personal norm (age to train) | 337 | 70% | 0.269 | 0.147 | 0.067 | -0.019 | 0.556 |
| **Personal norm (father's role)** | 3.71 | 1.15 | 0.153 | 0.063 | **0.015** | 0.030 | 0.276 |
| Village descriptive norm | 2.91 | 1.74 | 0.045 | 0.039 | 0.250 | -0.032 | 0.122 |
| Household injunctive norm | 4.54 | 1.07 | 0.082 | 0.067 | 0.220 | -0.049 | 0.214 |
| **Self-efficacy** | 4.00 | 0.87 | 0.219 | 0.095 | **0.022** | 0.032 | 0.406 |
| **Action control** | 3.92 | 1.41 | 0.699 | 0.050 | **<0.001** | 0.601 | 0.797 |
| Commitment | 3.80 | 1.26 | -0.090 | 0.061 | 0.139 | -0.210 | 0.029 |
| Intention | 4.51 | 0.91 | 0.020 | 0.086 | 0.816 | -0.148 | 0.188 |
| constant | | | -2.510 | 0.549 | <0.001 | -3.587 | -1.433 |

M = mean; SD = standard deviation; SE = standard error; CI = confidence interval

[+]N.B. Refer to S3 Table for the sample size for each specific variable

\*$R^2$ overall = 0.19; $R^2$ within villages = 0.14; $R^2$ between villages = 0.48; ICC = 0.01

\*\*$R^2$ overall = 0.47; $R^2$ within villages = 0.43; $R^2$ between villages = 0.42; ICC = 0.09

The psychosocial factors multivariate linear regression model for safe disposal intention identified household injunctive norm ($b$ = 0.154; 95% CI 0.045, 0.263), barrier planning ($b$ = 0.503; 95% CI 0.169, 0.837), and commitment ($b$ = 0.432; 95% CI 0.327, 0.537) as significant predictors (Table 4). Caregivers who perceived that most of their household members expected them to safely dispose of their child's feces had greater intention for the behavior. Caregivers who were strongly committed to always practicing safe disposal and had a plan for

how they would safely dispose of their child's feces when a barrier arose (i.e. water shortage) also had greater intention. The model explained 56% of the variance.

**Child latrine training.**    The mean values of the psychosocial factors for child latrine training were predominantly high, except for perceived vulnerability, proud attitudes towards latrine training, and village descriptive norm (Table 5). For perceived vulnerability, caregivers on average estimated the risk to their child's health from defecating outside was moderate to quite risky (M = 3.55). When it came to attitudinal factors, caregivers on average held strong negative attitudes towards child open defecation and strong positive attitudes towards latrine training. Caregivers believed it was a little or not at all beneficial for their child to defecate outside (M = 4.52) and were quite concerned for their child's safety when they did so (M = 4.14). In contrast, caregivers quite liked teaching their child how to use the latrine (M = 4.01), did not perceive it to be difficult (M = 4.24), and felt only a little irritated when they had to stop what they were doing to help their child use the latrine (M = 4.05). However, caregivers on average felt moderately proud when teaching their child how to use the latrine (M = 3.29). For norm factors, caregivers reported it was quite important to them personally to teach their child how to use the latrine (M = 4.06) and for the child's father to help with latrine training (M = 3.71), although slightly less so. Caregivers also perceived that most or all of their household members expected them to teach their child latrine use (M = 4.54). However, while 70% of caregivers believed a mother should start teaching her child how to use the latrine between 1 to 2 years old, caregivers estimated only some or half of the children in their village between 2 to 3 years old usually defecated in a latrine (M = 2.91). The mean self-efficacy score (M = 4.00) showed caregivers felt quite confident in their ability to teach their child how to use the latrine and to continue doing so in the face of challenges (e.g. during the night, when there is a water shortage, when child refuses). Lastly, caregivers had high mean values for their perceived action control over latrine training (M = 3.92), commitment (M = 3.80), and especially intentions (M = 4.51).

The psychosocial factors multilevel mixed effects regression model for latrine training intensity identified the following significant predictors (Table 5): perceived vulnerability of child open defecation (OD) (*b* = 0.098; 95% CI 0.011, 0.186), lack of benefit to child OD (*b* = 0.137; 95% CI 0.015, 0.258), safety concerns with child OD (*b* = -0.127; 95% CI -0.250, 00.005), liking latrine training (*b* = 0.238; 95% CI 0.099, 0.378), personal norm around father's role (*b* = 0.153; 95% CI 0.030, 0.276), self-efficacy (*b* = 0.219; 95% CI 0.032, 0.406), and action control (*b* = 0.699; 95% CI 0.601, 0.797). Caregivers who more strongly believed their child was at risk of becoming sick from OD and viewed child OD to be unbeneficial, more intensely taught their child how to use the latrine. In contrast, caregivers had lower latrine training intensity when they felt greater concern for their child's safety when the child defecated outside. Caregivers who liked teaching their child how to use the latrine and personally felt it was important for the child's father to help reported higher latrine training intensity. In addition, caregivers who felt confident in their ability to teach their child and who had greater action control over their training practice more intensely taught their child how to use the latrine. The model explained 47% of the variance overall, with similar amounts of variance explained among caregivers within the same village (43%) and between villages (42%).

## Discussion

We aimed to descriptively examine child latrine use and feces disposal practices, and subsequently identify contextual and psychosocial factors associated with two specific behaviors: caregiver safe disposal and child latrine training. Since caregiver safe disposal was found to be a rare practice—only 10% among children <5—we instead examined caregiver safe disposal

*intention*. With regard to contextual factors, we found caregivers had stronger behavioral intention to safely dispose when they came from wealthier households and had greater informational support, but less intention when the latrine was in/near the household compound. In contrast, caregivers more intensely practiced latrine training with their child when they themselves used the latrine for defecation, had a fully intact latrine, and greater instrumental support. With regard to psychosocial factors, caregivers had stronger intention to safely dispose when their households expected them to do so, they felt strongly committed to the behavior, and had a plan for what to do when faced with a water shortage barrier. For latrine training, caregivers more intensely taught their child how to use the latrine when they believed their child was at risk of becoming sick if they practiced OD; viewed child OD as unbeneficial; liked teaching their child; personally felt it was important for the child's father to help; felt confident in their ability to teach their child; and had greater action control over their training practice. Interestingly, caregivers put less effort into latrine training when they felt more concerned for their child's safety when the child defecated outside. These findings underscore the need to separately assess unique CFM practices in order to uncover the different behavioral factors at play. Moreover, our results offer a road map for practitioners working in similar settings on the types of behavior change strategies to consider in their CFM programming.

While many studies also showed household wealth and caregiver latrine use as associated with the broader term of "safe disposal" [10, 15–26], we parsed out these factors to reveal they are behaviorally specific: household wealth with caregiver safe disposal intention and caregiver latrine use with child latrine training. Household wealth may act as a proxy for a better quality latrine or exposure to more urban settings, such as a household member who works in a city, where caregiver safe disposal is more commonly practiced [9]. As for latrine training, it is intuitive that caregivers who use a latrine themselves would be more likely to then teach their child to use the latrine; this can also be viewed as a strategy in childrearing to model desired behaviors [43]. Interestingly, we also found caregivers have stronger intentions to safely dispose of their child's feces when their latrine is farther away from the household. Caregivers might prefer this greater distance at a specific child development stage: when their child is not yet ready to use the latrine but has started to eat adult foods and their feces now smell. However, since the vast majority of caregivers had a latrine in/near their household, it should be noted that only a few caregivers may have been driving this result. These findings emphasize the need to separately examine distinct CFM behaviors to reveal a clearer picture. Furthermore, for programming, it might prove harder to promote behavioral adoption of safe CFM practices in areas where households are poorer and there is less latrine use among adults.

We also uncovered different results by behavior for social support. Social support is an important resource in a person's social environment; it helps strengthen one's self-confidence and acts as a "buffer" against stressors [44, 45]. These differential findings are consistent with the stress-support matching hypothesis: a stressor must be matched with the type of support that is specifically needed [46]. The behaviors of safe disposal and child latrine training may be viewed in this light, as stressors in caregivers' daily lives among the myriad of other childcare and household responsibilities. Our results suggest caregivers desire informational support—advice and guidance—when it comes to safely disposing of their child's feces but desire instrumental support—acts of assistance—when it comes to latrine training. The desire for instrumental support aligns with the personal norm and self-efficacy psychosocial factors also found to be significantly associated with latrine training. Caregivers put greater effort into latrine training when it was personally important to them that the child's father help, potentially indicating that they were receiving this instrumental support from their husband. Caregivers also more intensively taught their child when they were more confident; in a separate examination using mediation analysis we show how instrumental support actually aids latrine training

through bolstering a caregiver's perceived self-efficacy [47]. As the WASH field recognizes how overburdened women are in taking up improved WASH practices, programmatic strategies that bolster support in the household or among other social networks could prove key.

One psychosocial factor associated with caregiver safe disposal intention related to injunctive norms. Descriptive norms are beliefs that others perform a certain behavior while injunctive norms are beliefs that others *expect* you to also perform this behavior [48]. When there is both a descriptive and injunctive norm in place, the behavior is deemed to be a "social norm" [48]. In this study we found caregivers held stronger safe disposal intentions when others in their household expected them to do this practice. Interestingly, we did not find a significant association with the village descriptive norm factor. Other qualitative studies have documented the role of both descriptive and injunctive norms around safe disposal. In urban Bangladesh, caregivers of children <3 years old explained that the common practice in their neighborhood (i.e. descriptive norm) was to throw child feces and diapers into open areas [33]. In western Kenya, caregivers of children <2 years old similarly emphasized it was a social norm to bury feces, dispose of wash water (from cleaning cloths/nappies) wherever was convenient, and there was no expectation for older siblings, who sometimes cared for their younger sibling, to practice safe disposal [28]. Taken together, these findings suggest interventions that aim to improve caregiver safe disposal should consider behavior change techniques that influence norms, such as public commitments or indicators of others' approval [49].

The other significant psychosocial factors for caregiver safe disposal intention were barrier planning and commitment, which are self-regulation factors. Self-regulation is about a person's ability to continue performing a behavior in spite of conflicting priorities, distractions, and other barriers [14]. As qualitative studies document, caregivers are busy with many tasks and find it difficult to incorporate new CFM practices [31, 33]. As such, along with norm techniques, CFM interventions focused on safe disposal can incorporate self-regulation techniques like goal setting and prompting caregivers to cope with barriers [49]. These results again suggest that strengthening informational support may be useful too: programs could engage a community mobilizer or family member to provide supportive guidance to caregivers on how to navigate different barriers with practicing safe disposal, such as water shortages, time and workload.

In contrast to caregiver safe disposal, child latrine training was associated with a wider variety of psychosocial factors. Along with norms and self-regulation, child latrine training was also associated with attitudes, self-efficacy, and risk perceptions; all of which require different types of behavior change techniques. For attitudes, caregivers more intensively taught their child how to use the latrine if they liked latrine training and viewed child open defecation as unbeneficial. Behavioral programs could include techniques that demonstrate latrine training as an enjoyable process, such as highlighting the positive emotions a caregiver might feel from training successes, and other techniques that illustrate the costs of the undesired behavior (i.e. child practicing open defecation). When it comes to improving caregivers' self-efficacy, some behavior change techniques include organizing social support, particularly instrumental support as already discussed, modeling or demonstrating the behavior, and provision of hardware [49]. There are mixed reviews on hardware like potties—some caregivers find them helpful while others do not [30, 33]. Novel hardware for latrine training may be needed, especially in contexts where children must learn to squat rather than sit when defecating. Lastly, caregivers more intensely taught their child how to use the latrine if they more strongly believed their child was at risk of becoming sick if they defecated outside. Behavioral programs could apply techniques that inform caregivers of the severe health risks that open defecation poses to their child.

Surprisingly, caregivers put less effort into latrine training when they felt more concerned for their child's safety when the child defecated outside. It could be that this risk perception serves as a proxy for when a caregiver perceives her child old enough to use the latrine. A few studies documented that caregivers perceive the latrine to be a dangerous place for small children, risking physical injury and illness [26, 34]. If a caregiver feels more concerned for her child's safety when they go for open defecation, she may also feel concerned about them using the latrine. However, this finding may have resulted from a type 1 error since we tested a large number of factors and the p-value was much higher compared to the other significant factors.

This study has some limitations. Firstly, as part of the eligibility criteria for the trial, all caregivers engaged had a household latrine and all villages had previously participated in the Gram Vikas MANTRA community-based water and sanitation intervention. As such, we caution against generalizing these findings to contexts where households do not have a strong enabling environment (i.e. access to a latrine and water) for practicing safe disposal and latrine training. Secondly, due to its uncommon practice, we were unable to directly examine the behavior of caregiver safe disposal and instead had to assess the behavioral precursor of *intention*. It is possible that different contextual and psychosocial factors are related to the behavior itself compared to intention; however, the two are viewed as closely linked and this approach has been used before [50]. Lastly, psychosocial factors related to child latrine training were examined solely from the perspective of the caregiver but different factors may be important from the child's perspective.

## Conclusions

As the water and sanitation field considers a more "transformative WASH" approach to reduce fecal exposure and achieve health gains [51], there is a need to better understand caregiver safe disposal and child latrine training. Our results exemplify it is critical to examine these behaviors separately so that their unique influencing factors are uncovered and more targeted behavioral strategies can be selected. However, while we urge the field to recognize these behaviors as distinct and to avoid the muddling term of "safe disposal," programmatically these behaviors should be addressed together as they are linked by child development. Caregivers first need to practice safe disposal but then as their child becomes developmentally ready they should transition to latrine training with their child. Our results showed it is around two years old when children start to use the latrine, which aligns with toilet training guidance from the Indian Academy of Pediatrics and others [52–54]. Practitioners can use the findings from this study to develop more holistic CFM programs that address both practices but apply specific behavior change techniques for each; tailor which content a caregiver receives based on her child's age; and also support those caregivers of younger children navigate the transition to latrine training as their child grows.

## Supporting information

**S1 Table. Caregiver and household demographics (N = 791)[+].** [+]The sample size varies by demographic variable, primarily due to respondents ending the survey early. One respondent said "don't know" for age. Here are the number of surveys with missing data by variable: 9 for drinking water, enclosed bathing area, two household latrines, and share latrine; 10 for latrine in use; 11 for education, religion, employment, and caste; 12 for household size; 14 for caregiver latrine use; 20 for latrine has piped water and location; 36 for latrine structure; and 52 missing for number of pits.
(DOCX)

**S2 Table. Descriptive statistics and bivariate regressions between caregiver safe disposal intention and predictor variables.**
(DOCX)

**S3 Table. Descriptive statistics and bivariate regressions between latrine training intensity and predictor variables.**
(DOCX)

## Acknowledgments

We would like to thank all of the participants who offered their time in taking the survey, especially when they were busy with caregiving tasks. We would also like to acknowledge Munmun DasMohapatra for translating the survey into Odia and her skillful training of the survey team, Indrajit Samal for providing invaluable logistical support throughout data collection, and Miriam Harter for her expertise and guidance on the piloting process of the psychosocial questionnaires. We are also grateful to the survey team who travelled far to reach all the study villages and had many long days of data collection.

## Author Contributions

**Conceptualization:** Gloria D. Sclar, Valerie Bauza, Alokananda Bisoyi, Thomas F. Clasen, Hans-Joachim Mosler.

**Data curation:** Gloria D. Sclar, Valerie Bauza.

**Formal analysis:** Gloria D. Sclar.

**Funding acquisition:** Gloria D. Sclar, Valerie Bauza, Thomas F. Clasen.

**Investigation:** Alokananda Bisoyi.

**Methodology:** Gloria D. Sclar, Valerie Bauza, Hans-Joachim Mosler.

**Project administration:** Gloria D. Sclar, Alokananda Bisoyi.

**Supervision:** Hans-Joachim Mosler.

**Visualization:** Gloria D. Sclar.

**Writing – original draft:** Gloria D. Sclar.

**Writing – review & editing:** Valerie Bauza, Alokananda Bisoyi, Thomas F. Clasen, Hans-Joachim Mosler.

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
