## [Decision Letter · Decision Letter 0]

7 Jun 2022

PONE-D-22-08579Contextual and Psychosocial Factors Influencing Caregiver Safe Disposal of Child Feces and Child Latrine Training in Rural Odisha, IndiaPLOS ONE

Dear Dr. Sclar,

Thank you for submitting your manuscript to PLOS ONE. After careful consideration, we feel that it has merit but does not fully meet PLOS ONE’s publication criteria as it currently stands. Therefore, we invite you to submit a revised version of the manuscript that addresses the points raised during the review process. Both reviewers have provided a couple of minor comments which should be addressed before acceptance.  I look forward to seeign the revised manuscript.

We look forward to receiving your revised manuscript.

Kind regards,

Alison Parker

Academic Editor

PLOS ONE

Journal Requirements:

“This work was supported, in whole or in part, by the Bill & Melinda Gates Foundation [INV-008967]. Under the grant conditions of the Foundation, a Creative Commons Attribution 4.0 Generic License has already been assigned to the Author Accepted Manuscript version that might arise from this submission.”

“TC received an award to fund this study from the Bill & Melinda Gates Foundation (https://www.gatesfoundation.org/; grant number = INV-008967). VB was funded in-part by the National Institute of Environmental Health Sciences, USA (https://www.niehs.nih.gov/; grant number = T32ES012870). The funders had no role in study design, data collection and analysis, decision to publish, or preparation of the manuscript.”

5. Please note that in order to use the direct billing option the corresponding author must be affiliated with the chosen institute. Please either amend your manuscript to change the affiliation or corresponding author, or email us at plosone@plos.org with a request to remove this option

6. We note that you have stated that you will provide repository information for your data at acceptance. Should your manuscript be accepted for publication, we will hold it until you provide the relevant accession numbers or DOIs necessary to access your data. If you wish to make changes to your Data Availability statement, please describe these changes in your cover letter and we will update your Data Availability statement to reflect the information you provide

7. Please include captions for your Supporting Information files at the end of your manuscript, and update any in-text citations to match accordingly. Please see our Supporting Information guidelines for more information: http://journals.plos.org/plosone/s/supporting-information

Reviewers' comments:

Reviewer's Responses to Questions

**Comments to the Author**

1. Is the manuscript technically sound, and do the data support the conclusions?

Reviewer #1: Yes

Reviewer #2: Yes

2. Has the statistical analysis been performed appropriately and rigorously? 

Reviewer #1: Yes

Reviewer #2: Yes

3. Have the authors made all data underlying the findings in their manuscript fully available?

Reviewer #1: Yes

Reviewer #2: Yes

4. Is the manuscript presented in an intelligible fashion and written in standard English?

Reviewer #1: Yes

Reviewer #2: Yes

5. Review Comments to the Author

Reviewer #1: The article makes a valuable contribution to available literature on sanitation and child faeces disposal more specifically. It is a well-designed study, technically correct, and clearly written. As such, I have only a few minor questions or suggestions as follows:

- Information on sample size may be included in the methods section.

- More information about data enumerators may be provided. Regarding the topic, sex of the enumerators may be relevant to report.

- Two versions of psychosocial variables sub-questionnaires were used based on whether a caregiver reported that their child is too young to use the latrine or not (p. 10). Is it possible to determine what age was on average considered as “old enough” for using latrine? If yes, I suggest reporting this information.

- The authors distinguish between contextual and psychosocial factors. I understand what the term psychosocial factors means. However, I am less sure that the term “contextual” (factors) is an appropriate label for all other factors. Could you clearly define/explain this label? For example, I would not refer to individual demographic or socioeconomic characteristics of caregivers as to contextual factors. Similarly, why social support variables were considered as contextual rather than psychosocial variables?

- I think that information stated on lines 252-253 was already provided earlier in the text and may be deleted.

- Were some of the variables centered? If yes, it should be mentioned in the methods section.

- Were assumptions on the data distributions checked for multilevel models and each level?

- Why were no explanatory variables considered at the level of villages? I was surprised that this possible contribution of multilevel modelling was not utilized. Consideration of such variables may also help with respect to the generalisability/transferability of findings.

Reviewer #2: This article is extremely well written. The literature review is informative and exemplary. Data are well analyzed and discussed.

There is one major issue not discussed, that warrants discussion. Given the data in Table 2 about the trend in child latrine use from age 0 to 59 months, would the authors recommend child ages for delivery of different intervention content. Given the schedule of routine immunization among children, are there specific ages when community health workers or facility-based health care providers might deliver specific content to the parents? My thinking is that there is an optimal child age for receipt of some intervention content, and programs are rarely nimble enough to provide the right information at the right time.

I would also like to see a brief discussion of the role of modeling of the behaviors / demonstration. Could more effort be invested in demonstration of the behaviors, and would that make a difference? Under what conditions might demonstrations be carried out?

6. PLOS authors have the option to publish the peer review history of their article (what does this mean?). If published, this will include your full peer review and any attached files.

Reviewer #1: No

Reviewer #2: No

---

## [Author Response · Author response to Decision Letter 0]

24 Jul 2022

Reviewer #1: The article makes a valuable contribution to available literature on sanitation and child faeces disposal more specifically. It is a well-designed study, technically correct, and clearly written. As such, I have only a few minor questions or suggestions as follows:

• Information on sample size may be included in the methods section.

Response by Authors

Thank you for this feedback. As noted in the paper, this study was part of a larger cluster RCT. So we did conduct a sample size calculation to determine the total number of clusters (i.e. villages) for the RCT which was 74 villages. This is described in detail in the protocol paper for the RCT. We added the following sentence in the methods to better direct readers to this sample size information (see Line 183):

Additional details about the trial study design, including sample size calculation to determine number of clusters (i.e. villages), are provided in Sclar, et al [35].

• More information about data enumerators may be provided. Regarding the topic, sex of the enumerators may be relevant to report.

Response by Authors

This is a very good point. We added information about the data enumerators including sex, language fluency, and the training that the team underwent. The following sentences were added to the section “Data collection procedure” (see Lines 197-202):

The data collection team consisted of 11 surveyors (10 women, 1 man) and a research manager (author AB), all of whom were fluent in the local language Odia. To ensure consistent data collection and accuracy of the survey tool, the team underwent a week-long training followed by several days of pilot-testing in non-trial villages. The survey was translated from English to Odia and checked by the data collection team during training, and further refined based on pilot-testing.

• Two versions of psychosocial variables sub-questionnaires were used based on whether a caregiver reported that their child is too young to use the latrine or not (p. 10). Is it possible to determine what age was on average considered as “old enough” for using latrine? If yes, I suggest reporting this information.

Response by Authors

Thank you for this feedback as this would indeed be useful information for the reader. We have now included a sentence that provides the average age for each analysis group – latrine training intensity group vs. safe disposal intention group (see Lines 425-427):

Children in the latrine training intensity group were on average about three years old (M = 38.87 months, SD = 11.49) while children in the safe disposal intention group were on average about one year old (M = 15.09 months, SD = 10.00).

As a reminder, caregivers who reported their child used the latrine or was old enough to use it were put in the “latrine training intensity” analysis group while caregivers who reported their child was not old enough to use latrine were put in the “safe disposal intention” analysis group. So we see that on average children 15 months old (or younger) were considered not old enough to use latrine. 

• The authors distinguish between contextual and psychosocial factors. I understand what the term psychosocial factors means. However, I am less sure that the term “contextual” (factors) is an appropriate label for all other factors. Could you clearly define/explain this label? For example, I would not refer to individual demographic or socioeconomic characteristics of caregivers as to contextual factors. Similarly, why social support variables were considered as contextual rather than psychosocial variables?

Response by Authors

Thank you for this feedback and we certainly appreciate this point. In the literature, there are several different ways in which the term “contextual factors” or “contextual determinants” gets used so it is important to be clear on how this term is being viewed in a given study. We explain what we mean by “contextual factors” compared to “psychosocial factors” in the introduction (see lines 104-116). In this paragraph of the introduction we explain that psychosocial factors cover cognitive or “mindset” elements that influence behavior (such as risk perceptions, attitudes, etc.) while contextual factors cover personal characteristics and aspects of the environment which may influence behavior. We specifically used these definitions because they separate those factors which can be addressed in an intervention (psychosocial factors – changing a person’s “mindset”) versus factors which you cannot really address in an intervention such as a person’s individual demographics or socioeconomic characteristics (hence, contextual factors – the elements that are outside the “mindset” or non-cognitive). 

We are viewing social support variables as contextual factors here, which we recognize are not always categorized in that way as you noted. However, here we specifically measure received social support rather than perceived support. So this is again not a cognitive element but an actual aspect of the person’s environment or “social context.” For example, some caregivers report that they do receive support with child feces management while others report they do not – this is a difference in the social context for the individual and is not about a difference in mindset (cognitive/psychosocial element). We summarize this concept in the following sentence in the introduction: “The environment is often viewed as physical elements that enable or impede behavior but it can also include social elements; for example, the availability of social support.”

We feel the introduction does an adequate job in clearly defining the “contextual factors” label and hence did not make any additional edits.

• I think that information stated on lines 252-253 was already provided earlier in the text and may be deleted.

Response by Authors

Thank you for noting this. The information is a little repetitive but we decided to keep it in because it is important information about how data on caregivers with multiple children was handled in different situations of the analysis.

• Were some of the variables centered? If yes, it should be mentioned in the methods section.

Response by Authors

No – none of the variables were centered.

• Were assumptions on the data distributions checked for multilevel models and each level?

Response by Authors

Thank you for bringing up the important point of assessing normality of the error distribution. For each of the four models, we generated histogram plots and normal Q-Q plots of the residuals. For the two multilevel mixed effects models we generated plots for both level 1 and level 2 residuals. Upon visual inspection, we found all plots indicated approximately normal distributions. We added the following sentence to the Methods section group (see Lines 352-354):

All variance inflation factors (VIFs) were low (<= 2.13), signifying no issues of multicollinearity in the regression models, and distribution of the errors was approximately normal upon visual inspection of histogram and quantile normal plots.

• Why were no explanatory variables considered at the level of villages? I was surprised that this possible contribution of multilevel modelling was not utilized. Consideration of such variables may also help with respect to the generalizability/transferability of findings.

Response by Authors

Thank you for this interesting point. Yes, we could have included contextual factors at the village level such as village size. However, we decided not to examine village-level contextual factors for two primary reasons:

1) First, identifying village-level contextual factors was not the goal of this research. We specifically wanted to understand factors which influence CFM behaviors regardless of which village the caregivers lived in. This was because we planned to use these findings to then design a behavior change intervention in collaboration with our implementing partner Gram Vikas (a local NGO). Gram Vikas specifically wanted to design a single intervention that they could implement across different types of villages they work in (so villages of different sizes, in different geographic locations across Ganjam and Gajapati districts, etc.). Therefore, we simply used multilevel models for the latrine training intensity analysis in order to account for/control village-level variation so that any significant factors that resulted were significant regardless of what village the caregiver lived in. 

2) Second, we also decided that including village-level contextual factors would be putting too much into the article. 

To make this clear, we included the following sentence in the methods (see Lines 349-352):

We did not examine village-level contextual factors in the multilevel model for latrine training intensity because the purpose of this study was to inform the design of a behavior change intervention that was meant to be implemented across the 74 different trial villages and thus not tailored to village context.

Reviewer #2: This article is extremely well written. The literature review is informative and exemplary. Data are well analyzed and discussed.

• There is one major issue not discussed, that warrants discussion. Given the data in Table 2 about the trend in child latrine use from age 0 to 59 months, would the authors recommend child ages for delivery of different intervention content. Given the schedule of routine immunization among children, are there specific ages when community health workers or facility-based health care providers might deliver specific content to the parents? My thinking is that there is an optimal child age for receipt of some intervention content, and programs are rarely nimble enough to provide the right information at the right time.

Response by Authors

This is an excellent point and something we originally had in the discussion but took out due to space. Yes, we do recommend interventions be tailored by child-age and take a child development lens whereby interventions focus on caregiver safe disposal for households with young toddlers (<2 years old) and focus on child toilet training for households with older toddlers (>= 2 years old). We revised the conclusion paragraph to include this age guidance and child development perspective (see Lines 743-755):

As the water and sanitation field considers a more “transformative WASH” approach to reduce fecal exposure and achieve health gains [51], there is a need to better understand caregiver safe disposal and child latrine training. Our results exemplify it is critical to examine these behaviors separately so that their unique influencing factors are uncovered and more targeted behavioral strategies can be selected. However, while we urge the field to recognize these behaviors as distinct and to avoid the muddling term of “safe disposal,” programmatically these behaviors should be addressed together as they are linked by child development. Caregivers first need to practice safe disposal but then as their child becomes developmentally ready they should transition to latrine training. Our results showed it is around two years old when children start to use the latrine, which aligns with toilet training guidance from the Indian Academy of Pediatrics and others [52-54]. Practitioners can use the findings from this study to develop more holistic CFM programs that address both practices but apply specific behavior change techniques for each; tailor which content a caregiver receives based on her child’s age; and also support those caregivers of younger children navigate the transition to latrine training as their child grows.

*Note that this revision added 3 citations to the reference list (53-54), which are all Academy of Pediatrics. 

• I would also like to see a brief discussion of the role of modeling of the behaviors / demonstration. Could more effort be invested in demonstration of the behaviors, and would that make a difference? Under what conditions might demonstrations be carried out?

Response by Authors

Thank you for bringing up this specific behavior change technique of modeling. In the RANAS approach, modeling/demoing a behavior is done to address behavioral factors related to “ability,” such as self-efficacy. When a behavior is modeled or demonstrated this helps a person feel more confident in their ability to perform the behavior themselves. Our psychosocial model of latrine training intensity showed caregivers more intensely taught their child to use the latrine when they felt confident in their ability to do so. As such, this suggests modeling techniques could be used in CFM programs to improve latrine training. We added this point – linking modeling as a behavior change technique to address self-efficacy – in the discussion (see Line 713):

When it comes to improving caregivers’ self-efficacy, some behavior change techniques include organizing social support, particularly instrumental support as already discussed, modeling or demonstrating the behavior, and provision of hardware [49].

---

## [Decision Letter · Decision Letter 1]

22 Aug 2022

Contextual and psychosocial factors influencing caregiver safe disposal of child feces and child latrine training in rural Odisha, India

PONE-D-22-08579R1

Dear Dr. Sclar,

We’re pleased to inform you that your manuscript has been judged scientifically suitable for publication and will be formally accepted for publication once it meets all outstanding technical requirements.

Kind regards,

Alison Parker

Academic Editor

PLOS ONE

Additional Editor Comments (optional):

Reviewers' comments:

Reviewer's Responses to Questions

**Comments to the Author**

1. If the authors have adequately addressed your comments raised in a previous round of review and you feel that this manuscript is now acceptable for publication, you may indicate that here to bypass the “Comments to the Author” section, enter your conflict of interest statement in the “Confidential to Editor” section, and submit your "Accept" recommendation.

Reviewer #1: All comments have been addressed

Reviewer #2: All comments have been addressed

2. Is the manuscript technically sound, and do the data support the conclusions?

Reviewer #1: Yes

Reviewer #2: Yes

3. Has the statistical analysis been performed appropriately and rigorously? 

Reviewer #1: Yes

Reviewer #2: Yes

4. Have the authors made all data underlying the findings in their manuscript fully available?

Reviewer #1: Yes

Reviewer #2: Yes

5. Is the manuscript presented in an intelligible fashion and written in standard English?

Reviewer #1: Yes

Reviewer #2: Yes

6. Review Comments to the Author

Reviewer #1: I had only minor suggestions in my previous assessment. All of them have been addressed or explained.

Reviewer #2: (No Response)

7. PLOS authors have the option to publish the peer review history of their article (what does this mean?). If published, this will include your full peer review and any attached files.

Reviewer #1: **Yes: **Josef Novotný

Reviewer #2: No

---

## [Editor Report · Acceptance letter]

30 Aug 2022

PONE-D-22-08579R1 

Contextual and psychosocial factors influencing caregiver safe disposal of child feces and child latrine training in rural Odisha, India 

Dear Dr. Sclar:

I'm pleased to inform you that your manuscript has been deemed suitable for publication in PLOS ONE. Congratulations! Your manuscript is now with our production department. 

Kind regards, 

on behalf of

Dr. Alison Parker 

Academic Editor

PLOS ONE